# Prospective Variation of Cytokine Trends during COVID-19: A Progressive Approach from Disease Onset until Outcome

**DOI:** 10.3390/ijms251910578

**Published:** 2024-10-01

**Authors:** Marina de Castro Deus, Ana Carolina Gadotti, Erika Sousa Dias, Júlia Bacarin Monte Alegre, Beatriz Akemi Kondo Van Spitzenbergen, Gabriela Bohnen Andrade, Sara Soares Tozoni, Rebecca Benicio Stocco, Marcia Olandoski, Felipe Francisco Bondan Tuon, Ricardo Aurino Pinho, Lucia de Noronha, Cristina Pellegrino Baena, Andrea Novais Moreno-Amaral

**Affiliations:** Programa de Pós-Graduação em Ciências da Saúde (PPGCS), Escola de Medicina, Pontifícia Universidade Católica do Paraná (PUCPR), Curitiba 80215-901, PR, Brazil; marina.deus@pucpr.edu.br (M.d.C.D.); ana_raixu3@hotmail.com (A.C.G.); erika.sousa@pucpr.edu.br (E.S.D.); julia.alegre@pucpr.edu.br (J.B.M.A.); beatriz.van@pucpr.edu.br (B.A.K.V.S.); gabriela.bohnen@pucpr.edu.br (G.B.A.); sara.tozoni@pucpr.edu.br (S.S.T.); rebecca.stocco@pucpr.edu.br (R.B.S.); marcia.olandoski@pucpr.br (M.O.); felipe.tuon@pucpr.br (F.F.B.T.); ricardo.pinho@pucpr.br (R.A.P.); lucia.noronha@pucpr.br (L.d.N.); cristina.baena@pucpr.br (C.P.B.)

**Keywords:** SARS-CoV-2, cytokine storm, severity, inflammation, COVID-19, hyperinflammation, death

## Abstract

COVID-19 is characterized by pronounced hypercytokinemia. The cytokine switch, marked by an imbalance between pro-inflammatory and anti-inflammatory cytokines, emerged as a focal point of investigation throughout the COVID-19 pandemic. However, the kinetics and temporal dynamics of cytokine release remain contradictory, making the development of new therapeutics difficult, especially in severe cases. This study collected serum samples from SARS-CoV-2 infected patients at 72 h intervals and monitored them for various cytokines at each timepoint until hospital discharge or death. Cytokine levels were analyzed based on time since symptom onset and patient outcomes. All cytokines studied prospectively were strong predictors of mortality, particularly IL-4 (AUC = 0.98) and IL-1β (AUC = 0.96). First-timepoint evaluations showed elevated cytokine levels in the mortality group (*p* < 0.001). Interestingly, IFN-γ levels decreased over time in the death group but increased in the survival group. Patients who died exhibited sustained levels of IL-1β and IL-4 and increased IL-6 levels over time. These findings suggest cytokine elevation is crucial in predicting COVID-19 mortality. The dynamic interplay between IFN-γ and IL-4 highlights the balance between Th1/Th2 immune responses and underscores IFN-γ as a powerful indicator of immune dysregulation throughout the infection.

## 1. Introduction

Patients hospitalized with COVID-19 between 2019 and 2021 may have experienced a dysregulated immune response known as a cytokine storm or hypercytokinemia, characterized by an excessive release of inflammatory cytokines [1]. This process involves an overproduction of pro-inflammatory cytokines [2], including IL-6, TNF-α, and IL-1β [3], with cytokine levels in severely affected patients significantly higher than in mild cases [4]. The immunopathogenesis of this response leads to lung damage, functional impairment, reduced lung capacity, and death [5]. Despite its clinical importance, molecular mechanisms driving cytokine storms remain under investigation due to the complex interplay of factors involved.

Cytokines are proteins secreted by immune cells for intercellular communication and signaling. They regulate angiogenesis, immune responses, inflammation, differentiation, and cell proliferation [6]. Elevated levels of IL-6 and TNF-α are strongly associated with COVID-19 severity. TNF-α amplifies the inflammatory response, while IL-6 contributes to tissue damage. IFN-γ is also correlated with disease severity, as individuals with severe COVID-19 exhibit elevated levels of pro-inflammatory cytokines, including IFN-γ, produced by CD4+ Th1 cells [7,8,9].

Interferons (IFNs) are crucial in innate immunity against viruses and other pathogens. IFN-γ activates the immune cascade, promoting the production of antiviral, antiproliferative, and immunomodulatory proteins, and is critical for viral clearance [6]. Similarly, IL-15 maintains innate immunity by balancing inflammatory cytokine production and the homeostasis of NK and CD8+ T cells, aiding in viral elimination [10].

TNF-α is pivotal in acute viral diseases and a key contributor to hypercytokinemia, while anti-inflammatory cytokines such as IL-10 and IL-4 regulate the inflammatory response [6]. Elevated levels of TNF-α and IFN-γ are associated with symptoms such as fever, chills, dizziness, fatigue, and headaches. They are also linked to conditions like cardiomyopathies, lung injury, and vascular leakage [11]. Despite its anti-inflammatory role, research suggests IL-10 also plays a role in fibrosis. Elevated tissue IL-10 expression correlates with increased collagen production and enhanced lung fibrocyte recruitment, contributing to disease progression [12,13].

IL-4 functions as an anti-inflammatory cytokine by inhibiting the production and activity of pro-inflammatory cytokines [14]. It exerts pleiotropic effects on a variety of target cells within both innate and adaptive immune responses [15] and is well-known for its role in B-cell co-stimulation [16] and promoting class switch recombination to IgE and IgG1 [17]. In a study by the COVID-19 study group at PUCPR, Motta Junior et al. (2020) found significant increases in mast cell density and IL-4 immunoreactivity in alveolar macrophages in lung biopsies from COVID-19 patients compared to H1N1 cases. This suggests that in severe SARS-CoV-2 infections, elevated IL-4 expression influences the proliferation and differentiation of mast cells recruited to the alveolar septum [18].

The hyperinflammatory response to COVID-19 is closely linked to disease severity and duration. Key mediators, including IL-6, IL-10, TNF-α, and IL-2, and chemokines like CCL2/MCP-1, CCL7, CXCL8/IL-8, CXCL10/IP-10, and CXCL11, play critical roles in the disease’s inflammatory spread [19]. Infected patients show elevated levels of IL-1β, IFN-γ, IP-10/CXCL10, and MCP-1/CCL2, contributing to activated T-helper-1 (Th1) cell responses. There is also evidence of increased T-helper-2 (Th2) cytokines, notably IL-4 and IL-10, which suppress inflammation [20].

SARS-CoV-2 interacts with host cells via the angiotensin-converting enzyme 2 (ACE2) receptor, triggering hypercytokinemia by dysregulating the ACE2/angiotensin II/AT1R axis and activating the complement system cascade, MAPK, and NF-κB pathway [6]. Elevated NF-κB activation produces pro-inflammatory cytokines, notably IL-6, inducing the cytokine storm observed in severe patients [21,22]. Virus binding to the ACE2 receptor also induces TGF-β expression [23,24], promoting differentiation of naive CD4+ T cells into T helper 17 (Th-17) cells, which produce cytokines like IL-17, GM-CSF, IL-21, and IL-2 [25]. IL-17 is directly associated with lung injury and COVID-19 severity [26], and it induces various pro-inflammatory cytokines and chemokines production [27,28]. This dysregulated immune response, coupled with alterations in the angiotensin II pathway, enhances hypercytokinemia, leading to significant lung injuries and systemic complications characteristic of severe COVID-19 cases [29].

To better understand the dynamics of cytokines involved in COVID-19 hypercytokinemia, well-designed, adequately powered prospective studies are essential. This deeper understanding is crucial for identifying risk profiles in hospitalized patients and contributing to the literature on COVID-19 progression.

## 2. Results

Out of 384 patients included in the study, 172 had only the first sample collected at admission, while 212 had two or more samples obtained at subsequent timepoints. The average number of samples collected in the survival group was 1.9 versus 4.0 in the death group. The longer the hospitalization, the more samples were collected (correlation 0.76). The average age of the patients in this study was 54.6 years (±16.6), with 115 patients (29.9%) being over 65 years old and 62.5% (*n* = 240) being male. The average BMI of the patients was 30 kg/m^2^ (±4.8), and 168 (49.7%) patients had a BMI greater than or equal to 30 kg/m^2^. The general clinical data of the patients included in the study are shown in Table 1. Also shown in Table 1 are the clinical characteristics of the study population when divided by outcome groups. The survival curve according to the length of hospital stay of the patients is shown in Figure 1.

### 2.1. Association between Cytokine Levels at Hospital Admission and Death

The first timepoint analysis measured cytokine levels from the first serum sample collected at hospital admission (*n* = 384). We observed that patients in the death group had significantly higher serum cytokine levels at the time of admission compared to the survival group (IL-6, IL-10, IFN-γ, IL-4, TNF-α, IL-15, and IL-1β: *p* < 0.001) (Table 2).

Cytokines IL-6, IL-4, IL-1β, and IFN-γ were significant predictors of death (*p* < 0.001). ROC curves were adjusted, and the area under the curve (AUC) was estimated (IL-4 AUC = 0.98; IL-1β AUC = 0.96; IL-6 AUC = 0.75; and IFN-γ AUC = 0.73) as shown in Figure 2. Comparing the cytokines pairwise regarding AUC, a significant difference was found between IL-4 and IL-6 (*p* < 0.001) and IL-4 and IFN-γ (*p* < 0.001), as well as between IL-1β and IL-6 (*p* < 0.001) and between IL-1β and IFN-γ (*p* < 0.001). No significant difference was found between IL-4 and IL-1β (*p* = 0.060) or between IL-6 and IFN-γ (*p* = 0.729). Therefore, the cytokines IL-4 and IL-1β showed the best performance in predicting a death outcome (Figure 2).

### 2.2. Prospective Analysis of Cytokine Levels between Outcome Groups

For prospective evaluation of the markers, only cases of patients with two or more serum samples collected at different timepoints were included in the analysis. A significant difference was observed in IL-4 and IL-1β levels between patients in the death group and survivors (*p* < 0.001). Patients in the death group presented higher levels of IL-4 and IL-1β at all timepoints evaluated, maintaining constant elevated levels throughout the course of infection (Figure 3C,D).

We observed higher IL-6 levels in patients with fatal outcomes throughout the infection (Figure 3B). Notably, there is a clear trend of increasing IL-6 levels in the death group, while in the survival group, the trend decreased. These opposite trends of marker values over time between outcome groups are confirmed by a significant interaction term between time after symptom onset and death (*p* < 0.001) (Figure 3B). The interaction term is used to determine whether the effect of one independent variable on the dependent variable differs across levels of another independent variable. The analysis of the interaction between time after symptom onset and death evaluates the comparison between death and survival patients to cytokine evolution, highlighting the interaction effect of time of infection based on mortality.

We observed that patients in the death group generally exhibited higher IFN-γ levels at the onset of the infection (Figure 3A). In contrast, survival patients showed a trend of increasing IFN-γ levels throughout the infection. Specifically, IFN-γ levels rose progressively in the survival group over time, while in the death group, IFN-γ levels were initially elevated but demonstrated a decreasing trend as the infection progressed (Figure 3A). We observed a significant difference in the interaction term between time after symptom onset and death (*p* < 0.001), indicating that the trend of IFN-γ evolution throughout the infection period varied between outcome groups. This finding confirms a trend of increasing IFN-γ levels over time in survival patients, while patients who succumbed to the infection exhibited a decrease in IFN-γ levels (Figure 3A).

## 3. Discussion

### 3.1. First Timepoint Analysis and Mortality

The analysis of first timepoint samples showed a significant elevation of all cytokines in the death group compared to the survival group. This finding suggests that, from the earliest stages of infection, namely at the time of hospital admission, patients who eventually died were already in a state of hypercytokinemia. Our results align with current knowledge of COVID-19 immunopathogenesis, highlighting the early increase in these markers and their correlation with disease mortality. A meta-analysis of 23 studies further supports this, showing that mean levels of IL-6, IL-8, IL-10, IL-2R, and TNF-α were significantly higher in severe cases compared to non-severe cases [2]. Analysis of cytokine concentrations in COVID-19 patients showed that, at the time of hospital admission, patients classified in the most critical severity group exhibited a dominant pro-inflammatory response with elevated levels of TNF-α, IL-6, IL-8, and IL-10, and high cytokine levels were associated with mortality [30]. These findings are similar to those observed by Del Valle et al. (2020), where samples from 1484 patients were measured at the time of hospital admission. The authors observed that elevated levels of IL-6, IL-8, and TNF-α were independent predictors of mortality in these patients [1]. In the present study, higher concentrations of IL-6, IL-10, and TNF-α, along with other evaluated markers, including IFN-γ, IL-1β, IL-15, and IL-4, were all associated to death outcome.

The seven pro-inflammatory cytokines (IL-6, IL-8, IL-15, IL-18, IL-27, IFN-γ, and TNF-α) and two anti-inflammatory cytokines (IL-1RA and IL-10) were elevated in COVID-19 patients compared to healthy controls [31]. Choreño-Parra et al. (2021) observed that pro-inflammatory cytokines (IFN-γ, IL-1β, IL-6, IL-9, IL-12, and CCL11) and anti-inflammatory cytokines (IL-4, IL-5, IL-10, and IL-13) were elevated in severely ill SARS-CoV-2 patients compared to pandemic influenza A (H1N1) patients. COVID-19 exhibited a distinct immunological profile characterized by increased levels of Th1 (IL-12 and IFN-γ) and Th2 (IL-4, IL-5, IL-10, and IL-13) cytokines, along with IL-1β and IL-6, among other markers [32]. These data suggest that SARS-CoV-2 induces an unbalanced inflammatory response different from the immune response in influenza. These findings indicate that SARS-CoV-2, but not H1N1 infection, induces both Th1 and Th2 responses. According to the literature, patients with COVID-19 show increased serum levels of GM-CSF and G-CSF compared to healthy volunteers, regardless of disease severity [33]. Elevated IL-17A and GM-CSF levels are linked to more severe cases, indicating dysregulated Th17 responses [33,34]. IL-17 overexpression in the lungs is associated with neutrophil infiltration and G-CSF-induced granulopoiesis [35]. Even after SARS-CoV-2 clearance, lung-resident Th17 cells (Trm17) maintain a pathogenic cytokine profile (IL-17A and GM-CSF), suggesting their role in driving severe hyperinflammation [34]. An increased frequency of GM-CSF/IFN-γ co-producing T cells has been observed in the blood of COVID-19 patients and is correlated with disease severity [36]. On the other hand, IL-17 further amplifies inflammation by recruiting monocytes, macrophages, and neutrophils and promoting the production of pro-inflammatory cytokines like IL-1β, IL-6, and TNF-α [36,37]. These findings and those observed in the present study suggest that the lack of regulation between the immune responses triggered after SARS-CoV-2 infection may be responsible for the immunological dysfunction observed during COVID-19.

We found that all prospectively evaluated markers, IFN-γ, IL-6, IL-1β, and IL-4, were significant predictors of death, with IL-4 and IL-1β showing the best performance for predicting the outcome of survival or death. The predictive power of IL-6 for mortality is extensively documented in the literature. Ozger et al. (2021) demonstrated that at various stages of infection, elevated levels of IL-6, IL-7, IL-10, IL-15, IL-27, IP-10, MCP-1, and GCSF were predictive of mortality, with IL-6, IL-10, IL-7, and GCSF showing the highest sensitivity and specificity [38]. Del Valle et al. (2020) also highlighted the predictive power of IL-6, as well as IL-8 and TNF-α [1]. Similarly, Ramos-González et al. (2024) demonstrated that IL-6 expression levels were associated with a higher risk of death, specifically showing that a 1% increase in cytokine expression was associated with a 7.3% increase in mortality risk [39]. Another study, evaluating 79 markers in 35 COVID-19 patients, identified IL-6, Eotaxin, IL-8, IL-1Ra, and MCP-1 as the strongest mortality predictors [40]. Similarly, a study by Basheer et al. (2022) involving 40 patients showed through multivariate analysis that IFN-γ and IL-10 were the most potent risk factors for mortality in COVID-19 [41]. In this context, our study adds the cytokines IL-4 and IL-1β to the markers with the highest sensitivity and specificity in predicting outcomes in our population.

In addition, in the present study, patient demographic and clinical characteristics mirrored those of the broader COVID-19 population, with shared features including age over 65 years (29.9%), obesity (49.7%), cardiometabolic diseases (13.3%), hypertension (39.5%), diabetes (22.9%), multiple comorbidities (53.1%), and admission SpO2 levels below 95% (66.4%). During the pandemic, factors such as age and obesity, as well as comorbidities such as hypertension, rheumatoid arthritis, and lung diseases, have been identified as predictors of COVID-19 severity and mortality [42,43,44]. These prognostic severity factors may be etiologically linked to COVID-19 pathogenesis, as they may regulate critical mediators of the host’s innate immune response. For example, obesity [45], inflammaging [46,47], cardiovascular disease [48], hypertension [49,50], and rheumatoid arthritis [51,52] are characterized by elevated concentrations of circulating cytokines, such as IL-1b, IL-6, IL-8, IL-10, IL-13, TNF-α, and IFN-γ. Cytokine overload related to the Th1 to Th2 shift in severe viral infection, when added to elevated cytokine levels from chronic conditions, may contribute to the CRS observed in COVID-19.

### 3.2. Prospective Cytokine Patterns

Prospective analyses showed that IFN-γ levels exhibit opposite patterns between the death and survival groups throughout the infection. In the death group, higher IFN-γ levels were observed at hospital admission, followed by a progressive decline before death. In contrast, patients in the survival group showed increasing IFN-γ levels over time. This trend of decreasing IFN-γ levels over time observed in the death group aligns with our previous study [9], which showed an initial elevation of IFN-γ in COVID-19 patients compared to healthy controls. However, this elevation did not persist, and IFN-γ levels decreased at the second evaluation timepoint. Notably, patients who maintained high IFN-γ levels at the second timepoint showed a significant association with a poorer prognosis, contrasting with the observations in the present study. Another study reported that severe patients exhibited high IFN-γ levels at the beginning of infection, with a rapid decline over time, while moderate patients showed less significant but stable IFN responses throughout the infection [53].

Numerous studies have described the ability of the COVID-19 virus to suppress the IFN-γ response at the onset of infection [53,54]. This suppression leads to an accumulation of monocytes and macrophages [55,56], resulting in increased cytokine production in the lungs [57], including IFN-γ. Other studies highlight the capacity of NK cells to produce IFN-γ before the specific Th1 response [58], explaining the observed increase in IFN-γ at the beginning of infection. However, reports on the role and expression levels of IFN-γ in COVID-19 immunopathogenesis are inconsistent. Some studies report intensified IFN responses [53,59,60], while others indicate reduced responses [61,62] in severe cases at infection onset, with insufficient data on the prospective trajectory of this marker throughout infection. The action of IFN-γ is crucial for viral clearance. A robust T-cell response in mild COVID-19 cases enables rapid and efficient viral clearance. Conversely, patients who died exhibited a weak cellular response and low IFN-γ secretion 26 days after symptom onset [63], similar to our findings. Our results underscore the inefficient immune response in fatal COVID-19 cases. These patients initially produce high IFN-γ levels but cannot sustain this production, impairing their capacity to combat the infection over time.

Our prospective analysis of IL-6 showed an increasing trend in marker levels in the death group. In contrast, survival patients could regulate inflammation levels, demonstrating a decreasing trend in IL-6 over time. IL-1β exhibited a similar pattern, with elevated levels in the death group throughout the infection, though these levels remained stable without a clear trend of increase or decrease. Our findings are consistent with other prospective studies that observed an increase in IL-6 in severe patients [1,64]. Ling et al. (2021) found that severe patients had progressively higher IL-6 levels during the infection, while IL-1β levels were initially lower but increased at the second evaluation timepoint [64]. Similarly, Yudhawati (2022) reported that IL-1β concentrations tended to decrease in the moderate group but increased in the severe group [65]. However, other prospective studies did not observe the same increasing trend. Lucas et al. (2020) described severe patients with higher levels of IL-6 and IL-1β throughout the infection, but no increasing trend was noted over time [59].

During SARS-CoV-2 infection, the interaction between the virus and its cellular receptor (ACE2) activates the NF-κB pathway, resulting in the production of IL-6, TNF-α, IL-1β, and IL-10 [25,66]. Elevated IL-6 concentrations have been identified as an important predictor of mortality in COVID-19 [1,67], as also shown in the present study. Severe patients also exhibit high levels of IL-1β, associated with thrombi and Acute Respiratory Distress Syndrome (ARDS) [68]. The virus influences the maturation and activation of IL-1β, which, in turn, can stimulate the production of other inflammatory cytokines like IL-6 [69,70,71], thereby accelerating inflammation, contributing to hypercytokinemia, and leading to a poorer prognosis [36].

We found that IL-4 levels were consistently elevated over time in the death group compared to the survival group. In our previous study [9], IL-4 was elevated in the later stages of infection (>10 days) and was not associated with mortality. However, in the present study, IL-4 levels remained high throughout the infection in patients in the death group, making IL-4 a significant predictor of mortality. Another study also demonstrated that increased IL-4 expression was correlated not only with mortality but also with the development of respiratory failure and acute renal failure in COVID-19 patients [72]. These findings differ from other studies, which found lower IL-4 levels in severe patients [73] and in those who died and were admitted to the ICU at the time of hospital admission [74].

The literature varies on reports on IL-4 expression during COVID-19, similar to the observations for IFN-γ. IL-4 is an anti-inflammatory cytokine involved in the Th2 immune response and interferes with the Th1 immune response [75]. Conversely, IFN-γ, part of the Th1 response, inhibits Th2 mediators production [76]. Thus, IL-4 is generally not studied simultaneously with IFN-γ and IL-6 to better characterize the Th2 and Th1 immune responses [9]. The present study highlights the dynamic interplay between IFN-γ and IL-4 between Th1 and Th2 immune responses, with IFN-γ as a critical indicator of immune dysregulation during COVID-19. This balance is crucial for an effective immune response, where the predominance of one type over the other can lead to adverse clinical outcomes. Our findings suggest that the inability to maintain adequate levels of IFN-γ is associated with a worse prognosis in patients with COVID-19. Even with the pandemic under control, these insights remain relevant for understanding its immunopathogenesis and future research on therapeutic strategies in viral diseases and inflammatory conditions.

## 4. Materials and Methods

### 4.1. Study Design and Patients

This study was approved by the National Research Ethics Committee (Conselho Nacional de Ética em Pesquisa—CONEP), protocol number 3.944.734/2020 Serum samples were collected from patients hospitalized at Marcelino Champagnat Hospital in Curitiba, PR, Brazil. The inclusion criteria were patients over 18 years old with a positive RT-PCR test for COVID-19, non-transplanted, immunocompetent, and who did not receive tocilizumab during hospitalization. Of 442 hospitalized patients with COVID-19 symptoms, 384 were included in the study. The date of symptom onset was reported by the patient at hospital admission and recorded for prospective analysis. Clinical data were collected from the hospital’s information system, and patients were grouped based on their outcomes (death or survival) for analysis.

### 4.2. Sample Processing and Cytokine Measurements

The first sample was obtained at hospital admission and subsequently collected every 72 h until hospital discharge or death. Blood samples were taken in SST II Tubes (BD, Biosciences, Franklin Lakes, NJ, USA) and centrifuged for 15 min at 3000 rpm The serum was collected and stored at −80 °C until analysis. According to the manufacturer’s instructions, serum cytokines TNF-α, IFN-γ, IL-1β, IL-15, IL-6, IL-10, and IL-4 were quantified using commercial ELISA kits (ImmunoTools, Friesoythe, Germany). Results were read using the VersaMax™ Microplate Reader (Molecular Devices, San Jose, CA, USA) at the wavelength specified by the kit.

### 4.3. Sample Size Calculation

The sample size calculation was conducted to detect a significant interaction between group and time (evolution of discharge and death groups throughout hospitalization) using a general linear mixed model (GLMM). Separate calculations were performed for each cytokine evaluated, with the final sample size determined by the maximum value required for any cytokine. Initial estimates of means, standard deviations, and correlation coefficients between time points and effect size were derived from data on the first 15 patients with complete follow-up until discharge or death. A minimum sample size of 210 patients was needed to detect a significant interaction, assuming a death-to-survival ratio of 1:4, with 90% statistical power and a significance level of 5%. The power analysis was performed using the General Linear Mixed Model Power and Sample Size (GLIMMPSE) software, version 3, available at https://samplesizeshop.org/glimmpse-power-software/ (accessed on 5 September 2020) [77].

### 4.4. Statistical Analysis

Data were organized in Excel^®^ (https://www.microsoft.com/en-us/microsoft-365/excel) and analyzed using Stata/SE v.14.1 (Stata Corp. LP, College Station, TX, USA). Quantitative variables were described using the mean, standard deviation, median, minimum, maximum, and interquartile range (IQR). Categorical variables were described using absolute and relative frequencies. Non-normally distributed variables were log-transformed. Group comparisons (death vs. survival) regarding cytokine expression were performed using Student’s t-test for independent samples. Mixed-effects linear regression models were employed to analyze the evolution of cytokine levels over time, accounting for group (death or survival), symptom duration, and the interaction between group and time. Residual plots revealed no loss of homoscedasticity or normality. Cytokine levels in prospective analyses were expressed as log pg/mL with confidence intervals. Survival times (event of death or hospital discharge) were described using a Kaplan–Meier curve with 95% confidence intervals. ROC curves were generated, and the area under the curve (AUC) was estimated to evaluate the performance of cytokines in discriminating between survival and death outcomes. Statistical significance was defined as a *p*-value of <0.05.

## 5. Conclusions

The present study confirmed that hypercytokinemia is prominent in patients with a fatal outcome. This excessive immune response, marked by the simultaneous elevation of numerous cytokines at the same stage of infection, is significantly associated with mortality and can serve as a predictor. Early control of the cytokine storm through therapies such as immunomodulators and cytokine antagonists is crucial for improving the survival rate of COVID-19 patients [29].

A significant limitation of this study was the inability to isolate the effects of factors such as obesity, medication, and hospitalization complications like bacterial infections and sepsis on the cytokine measurements. Bacterial infections can trigger secondary immune responses, altering cytokine expression and leading to increased inflammatory mediators and hypercytokinemia, complicating the interpretation of cytokine elevations associated with COVID-19. Treatments such as corticosteroids and antibiotics, including azithromycin, can further influence cytokine profiles in patients with prolonged hospitalization, making it difficult to determine the specific impact of COVID-19 on inflammatory markers. Additionally, 49.7% of the cohort was affected by obesity, known to exacerbate inflammatory cytokine expression, as excess adipose tissue secretes elevated levels of pro-inflammatory adipokines like TNF-α and IL-6. These factors collectively affect cytokine dynamics throughout the disease. However, septicemia was documented as the primary cause of death in only 23 patients within the hospital’s database. Furthermore, our study did not include healthy individuals for a comparative analysis of cytokine levels.

Despite the inability to isolate the effects of obesity, medications, and hospitalization complications on cytokine measurements, our study highlights the significance of cytokine levels in predicting mortality among COVID-19 patients. By analyzing inflammatory mediators, we can enhance the early identification of critically ill patients, facilitating more effective clinical decision-making in the future.

In summary, our research highlights IFN-γ as a crucial cytokine in COVID-19 progression and severity, revealing distinct profiles between patients who survive and those who do not. The variability in the literature concerning IFN-γ underscores the significance of our study, which assessed IFN-γ at multiple infection stages and examined the dynamic release patterns of IL-4, IL-1β, and IL-6. These findings offer critical clinical insights and have a lasting impact even with the pandemic under control. They underscore the need for continued research into cytokine profiles, which could enhance our understanding of immune responses in COVID-19 and other viral infections, ultimately guiding the development of more effective, targeted therapies.

## Figures and Tables

**Figure 1 ijms-25-10578-f001:**
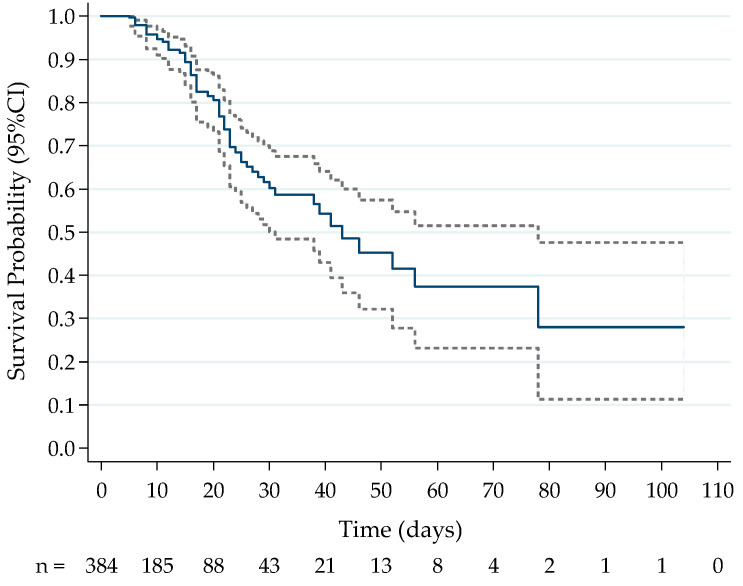
Kaplan–Meier curve of survival probability and 95% confidence interval (event of death or hospital discharge) of the patients included in the study, *n* = 384. The solid line describes the probability of survival. The dashed lines represent the 95% confidence interval. Time expressed in days of hospitalization.

**Figure 2 ijms-25-10578-f002:**
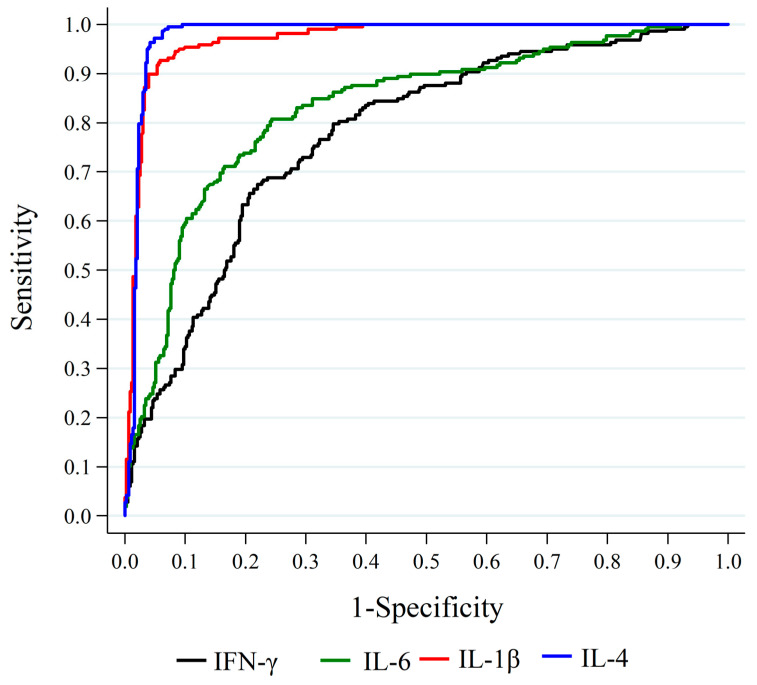
ROC curves with estimated area under the curve (AUC) for cytokines IL-4, AUC = 0.98; IL-1β, AUC = 0.96; IL-6, AUC = 0.75; IFN-γ, AUC = 0.73.

**Figure 3 ijms-25-10578-f003:**
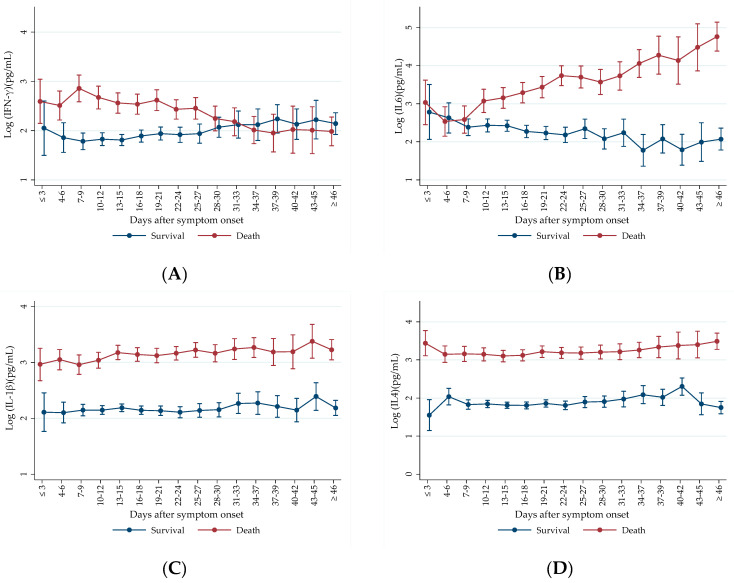
Comparison of predicted values from the logarithmic model for cases with at least two timepoint samples between the death and survival groups, based on the number of days since symptom onset. Cytokine values are expressed in log pg/mL with a confidence interval. ‘Survival’ refers to patients who were discharged from the hospital, and ‘Death’ refers to patients who had a fatal outcome. (**A**) Comparison of predicted values from the logarithmic model for cases with at least two IFN-γ samples (*n* = 212), interaction value between symptom duration and death: *p* < 0.001. (**B**) Comparison of predicted values from the logarithmic model for cases with at least two IL-6 samples (*n* = 212), interaction value between symptom duration and death: *p* < 0.001. (**C**) Comparison of predicted values from the logarithmic model for cases with at least two IL-1β samples (*n* = 211), interaction value between symptom duration and death: *p* = 0.108. The difference in marker levels between outcome groups: *p* < 0.001. (**D**) Comparison of predicted values from the logarithmic model for cases with at least two IL-4 samples (*n* = 211), interaction value between symptom duration and death: *p* = 0.277. Difference in marker levels between outcome groups: *p* < 0.001.

**Table 1 ijms-25-10578-t001:** Clinical data of the patients included in the study.

Clinical Variables	All Patients *n* = 384	Outcome	
		Survival *n* = 324	Death *n* = 60
Age (years)	54.6 ± 16.6	51.7 ± 15.1	70 ± 15.5 **
Age ≥ 65 years	115 (29.9%)	71 (21.9%)	44 (73.3%) **
Males	240 (62.5%)	205 (63.3%)	35 (58.3%)
BMI (kg/m^2^)	30 ± 4.8	30.1 ± 4.9	29.2 ± 4.8
BMI ≥ 30 kg/m^2^	168 (49.7%)	147 (51.2%)	21 (41.2%)
Temperature at admission (°C)	36.5 ± 0.6	36.5 ± 0.6	36.5 ± 0.6
Saturation at admission (%)	91.6 ± 5.4	92.3 ± 4.3	87.8 ± 8.2 **
Saturation < 95%	255 (66.4%)	207 (63.9%)	48 (80%) *
Systolic blood pressure at admission	128.7 ± 19.9	128.6 ± 18.8	128.7 ± 25.2
Diastolic blood pressure at admission	77.8 ± 15.3	78.7 ± 15.4	73.2 ± 14.1 *
Heart rate at admission	92.9 ± 18.4	93.4 ± 18.6	90.4 ± 16.9
Respiratory rate at admission	22.1 ± 5.3	21.5 ± 5.1	25.2 ± 5.6 **
Systemic arterial hypertension (SAH)	152 (39.6%)	114 (35.2%)	38 (63.3%) **
Diabetes mellitus	88 (22.9%)	68 (21%)	20 (33.3%) *
Dyslipidemia	81 (21.1%)	61 (18.8%)	20 (33.3%) *
Cardio-metabolic disease	51 (13.3%)	29 (9%)	22 (36.7%) **
Orotracheal intubation	115 (29.9%)	59 (18.2%)	56 (93.3%) **
Rheumatic disease	14 (3.6%)	10 (3.1%)	4 (6.7%)
Respiratory disease	42 (10.9%)	29 (9.0%)	13 (21.7%) **
2 or more pre-existing comorbidities	204 (53.1%)	157 (48.5%)	47 (78.3%) **
4 or more pre-existing comorbidities	69 (18%)	43 (13.3%)	26 (43.3%) **

Results are described as mean ± standard deviation or median (minimum–maximum) for quantitative variables or frequency (percentage) for categorical variables. Student’s *t*-test for independent samples or Mann–Whitney non-parametric test (quantitative variables); Fisher’s exact test (categorical variables). * *p* < 0.05; ** *p* < 0.001.

**Table 2 ijms-25-10578-t002:** Cytokine levels between outcome groups at hospital admission.

Cytokine	Outcome	*n*	Median	Minimum	Maximum	IQR	*p* *
Log (IL-6) (pg/mL)	Survival	323	2.29	0.69	4.60	1.05	
	Death	60	3.27	1.40	5.95	1.05	<0.001
Log (IL-10) (pg/mL)	Survival	312	1.86	0.67	5.21	0.49	
	Death	58	2.76	0.70	5.25	1.61	<0.001
Log (IFN-γ) (pg/mL)	Survival	322	1.93	−0.30	3.28	0.87	
	Death	60	2.37	1.24	4.44	0.59	<0.001
Log (IL-4) (pg/mL)	Survival	321	1.76	0.34	3.88	0.57	
	Death	60	3.21	2.50	3.54	0.40	<0.001
Log (TNF-α) (pg/mL)	Survival	322	2.01	0.78	4.05	0.56	
	Death	60	3.29	0.54	4.40	0.90	<0.001
Log (IL-15) (pg/mL)	Survival	314	1.87	0.07	3.63	0.83	
	Death	55	2.40	1.87	3.54	0.35	<0.001
Log (IL-1β) (pg/mL)	Survival	309	2.07	0.93	3.93	0.42	
	Death	53	3.12	2.23	4.11	0.34	<0.001

Values expressed as median, minimum, and maximum. * Student’s *t*-test for independent samples, *p* < 0.05 (data subjected to logarithmic transformation). IQR, inter-quartile range (difference between 1st and 3rd quartiles).

## Data Availability

The data supporting this study’s findings are available from the corresponding author (Andréa Novais Moreno-Amaral) upon reasonable request.

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
