# Peer review of "Prospective Variation of Cytokine Trends during COVID-19: A Progressive Approach from Disease Onset until Outcome"

_ijms, 2024, doi:10.3390/ijms251910578_

Round 1

Reviewer 1 Report

Comments and Suggestions for Authors

The manuscript “Prospective variation of cytokine trends during COVID-19: a progressive approach from disease onset until outcome” by de Castro Deus et al. provided a longitudinal observation on hypercytokinemia in COVID-19 patients and the co-relationship between the cytokine levels and the disease outcomes, which is intriguing. Although there are many overlaps with early studies, this study revealed more parameters that could be used to predict survival or death. The information is important for future research on therapeutic strategies.

I have a few concerns about the study:

1.      There are no measures of IL-17, GM-CSF, and G-CSF. Although Th17 responses may not be important in antiviral immunity, they may join the inflammatory reactions/hypercytokinemia.    

2.      The quality of figures is very poor, which should be improved.

3.      Line 60-61, the description of the role of T cells in immunoglobulin classes is not accurate. A more updated reference should be cited.

Author Response

Comments 1: There are no measures of IL-17, GM-CSF, and G-CSF. Although Th17 responses may not be important in antiviral immunity, they may join the inflammatory reactions/hypercytokinemia.

Response 1: Thank you for bringing this to our attention. In developing this COVID-19 project, our initial research focused on the Th1/Th2 axis of immune responses during SARS-CoV-2 infection. While our study did not include direct measurements of IL-17, GM-CSF, or G-CSF, emerging literature during our research highlighted a potential association between elevated levels of these cytokines and increased disease severity. Although Th17 responses are not classically linked to antiviral immunity, they can contribute to hypercytokinemia and amplify inflammatory responses. Given this, we have addressed the possible role of Th17 responses and their associated cytokines (IL-17, GM-CSF, G-CSF) in the discussion section of our manuscript (Lines 203-215). This addition provides a more comprehensive view of the immune mechanisms that may exacerbate COVID-19 severity.

Comments 2: The quality of figures is very poor, which should be improved.

Response 2: We thank you for highlighting this issue. All figures were carefully reviewed in the revised paper, and the resolution was improved to ensure better quality and facilitate clear visualization of the results presented.

Comments 3: Line 60-61, the description of the role of T cells in immunoglobulin classes is not accurate. A more updated reference should be cited.

Response 3: We appreciate the reviewer's insightful comment and have revised the sentence to enhance accuracy. The updated paragraph can be found in Lines 58-62.

Reviewer 2 Report

Comments and Suggestions for Authors

Dear Authors,

I have read with interest your manuscript and I think that it is very well written and the topic is of interest.

I have few questions for you:

1) In Table 1 you have not reported data concerning the presence of rheumatic diseases, could you excluded these?

 2) in Table 1 you did not describe the presence of patients with respiratory diseases, I think that it is not possible for epidemiological data

3) Please revise discussion considering the data od Table 1 and 2, in fact the presence of these manifestations could increase the basal levels of ILs.

4) In table 2 Please add the normal range of cytokines

5) In methods, please add the power calculation 

Comments on the Quality of English Language

None

Author Response

Comments 1: In Table 1 you have not reported data concerning the presence of rheumatic diseases, could you excluded these?

Response 1: Thank you for raising this point. The primary focus of our study was to examine the relationship between COVID-19 severity and specific inflammatory biomarkers throughout the infection until the outcome. Although the presence of rheumatic diseases was not excluded, it was not initially reported due to the limited number of participants with this condition (3.6% of the total population) (n=14). We have included this information in Table 1 and discussed the rheumatic, respiratory diseases, and other comorbidities that could influence baseline IL levels (Lines 234-247). Future studies may benefit from a more detailed analysis of these subpopulations to explore potential associations.

Comments 2: In Table 1 you did not describe the presence of patients with respiratory diseases, I think that it is not possible for epidemiological data.

Response 2: Thank you for your observation. We acknowledge the importance of including respiratory diseases in epidemiological data, especially given their potential impact on inflammatory markers. In our study, patients with respiratory diseases were not initially highlighted in Table 1 due to the study's primary focus on inflammatory biomarkers. However, we agree that reporting these comorbidities is essential for a comprehensive understanding, and we have now added this information to Table 1 for clarity and completeness. To improve our discussion, a new sentence addressed how previous comorbidities can be related to cytokine release syndrome in COVID-19 (Lines 235-248).

Comments 3: Please revise discussion considering the data od Table 1 and 2, in fact the presence of these manifestations could increase the basal levels of ILs.

Answer 3: We appreciate your comment. We agree that the specific manifestations, presented in Tables 1 and 2, may contribute to elevated baseline IL levels, and this has already been addressed as a limitation of the study (Lines 379-392). However, given the relevance of this topic in cytokine release syndrome, we used data from the literature to discuss how comorbidities, such as respiratory and rheumatic diseases, may influence the observed inflammatory profile (Lines 235-248). This adjustment reinforces the potential impact of underlying conditions on IL levels and aligns our findings with the broader context of inflammatory responses in patients with COVID-19.

Comments 4: In table 2 Please add the normal range of cytokines.

Response: Thank you for this valuable suggestion. However, our study did not include a negative control group, which limits our ability to provide a reliable "normal" reference range for the cytokines. Establishing accurate reference values requires data from a control group that undergoes the same experimental conditions—using identical antibodies, protocols, sample types, and handling procedures. Without such a reference group, including normal ranges could be misleading.

Furthermore, reported cytokine ranges in the literature can vary significantly, even when measured with the same technique (e.g., ELISA) and expressed in the same units (pg/ml). Differences arise from variations in experimental conditions, population demographics, and the assay sensitivity used. Other studies might employ different methodologies, such as Multiplex bead-based immunoassays, which are not directly comparable to ELISA due to distinct detection mechanisms and sensitivity profiles. As such, reporting a general "normal range" risks introducing inaccuracies in our data interpretation.

To address this, we have added this limitation of our study to the discussion section, highlighting these challenges and explaining the complexities of establishing universal reference ranges for cytokines (Lines 391-392).

Comments 5: In methods, please add the power calculation.

Response: Thank you for your valuable comment. We have added the power calculation to the Methods section (Lines 342-355).

Round 2

Reviewer 2 Report

Comments and Suggestions for Authors

none

Comments on the Quality of English Language

none